# Healthcare experiences of perpetrators of domestic violence and abuse: a systematic review and meta-synthesis

Marilia A Calcia ![ORCID],[1] Simran Bedi,[1] Louise M Howard,[1] Heidi Lempp,[2] Sian Oram[1]

¹Section of Women's Mental Health, Department of Health Service and Population Research, Institute of Psychiatry, Psychology and Neuroscience, King's College London, London, UK
²Centre for Rheumatic Diseases, Department of Inflammation Biology, Faculty of Life Sciences and Medicine, King's College London, London, UK

**Correspondence to**
Dr Marilia A Calcia;
marilia.a.calcia@kcl.ac.uk

## ABSTRACT

**Objectives** Domestic violence and abuse (DVA) is highly prevalent, with severe adverse consequences to the health and well-being of survivors. There is a smaller evidence base on the health of DVA perpetrators and their engagement with healthcare services. This review examines the experiences of perpetrators of DVA of accessing healthcare services and the barriers and facilitators to their disclosure of abusive behaviours in these settings.

**Design** A systematic review and meta-synthesis of qualitative studies.

**Data sources** A systematic search was conducted in Cochrane, MEDLINE, Embase, PsycINFO, HMIC, BNID, CINAHL, ASSIA, IBSS, SSCI (peer-reviewed literature) and NDLTD, OpenGrey and SCIE Online (grey literature). Each database was searched from its start date to 15 March 2020. Eligibility criteria required that studies used qualitative or mixed methods to report on the experiences of healthcare use by perpetrators of DVA. A meta-ethnographic method was used to analyse the extracted data.

**Results** Of 30,663 papers identified, six studies (n=125 participants; 124 men, 1 woman) met the inclusion criteria. Barriers to disclosure of DVA to healthcare staff included perpetrators' negative emotions and attitudes towards their abusive behaviours; fear of consequences of disclosure; and lack of trust in healthcare services' ability to address DVA. Facilitators of disclosure of DVA and engagement with healthcare services were experiencing social consequences of abusive behaviours; feeling listened to by healthcare professionals; and offers of emotional and practical support for relationship problems by healthcare staff.

**Conclusions** DVA perpetration is a complex issue with multiple barriers to healthcare engagement and disclosure. However, healthcare services can create positive conditions for the engagement of individuals who perpetrate abuse.

**PROSPERO registration number** CRD42017073818.

## Strengths and limitations of this study

► This study provides a systematic review of the healthcare experiences of domestic violence and abuse (DVA) perpetrators, including facilitators and barriers to disclosure of abusive behaviours to healthcare professionals.
► Robust procedures for systematic reviewing and quality assessment were adopted, in line with Preferred Reporting Items for Systematic Reviews and Meta-Analyses (PRISMA) reporting guidelines.
► The number of studies identified was small and some healthcare settings, for example, emergency department, were not represented in the study sample.
► This review identified studies conducted in high-income countries and most participants were men who perpetrated intimate partner violence; the findings may, therefore, be less applicable to other settings such as low-income and middle-income countries, or other populations such as perpetrators of family violence or female perpetrators of DVA.

victims of homicide have been killed by an intimate partner.[1] In the UK, the definition of domestic violence includes family violence (FV) as well as IPV. DVA is defined in the UK Domestic Abuse Bill 2020 as abusive behaviour (which may be physically, sexually, emotionally, psychologically or economically abusive or violent, threatening, controlling or coercive) by one person to another, when both are aged 16 or over and are 'personally connected'. The term personally connected refers to two people who are, or have been, in an intimate personal relationship, marriage or civil partnership, where each has parental responsibility to a child, or who are relatives.[2] In the year ending March 2018, an estimated 2 million people experienced DVA in England and Wales.[3] Female victims outnumbered male victims by approximately 2:1.

DVA has a range of impacts on survivors' health. Up to 42% of women affected by IPV have reported injuries as a consequence of it.[4] Chronic pain, gastrointestinal problems,

## INTRODUCTION

Domestic violence and abuse (DVA) are highly prevalent worldwide. According to the WHO, 27% of women aged 15–49 years who have been in an intimate relationship have experienced physical or sexual intimate partner violence (IPV) and 38% of female

gynaecological problems, depression, anxiety and other mental disorders are also known to be associated with DVA.[5–7] Women who have experienced IPV are at higher risk of having a low-birth-weight baby, abortions, depression and of acquiring HIV compared to women with no history of IPV.[4] The adverse outcomes of DVA also affect children; children in families where DVA occurs are at increased risk of developing adverse behavioural, emotional and cognitive outcomes.[8–10]

Less evidence exists regarding the health of DVA perpetrators, of which most relates to mental health. Systematic review evidence suggested both men and women with mental disorders are more likely to have a lifetime history of physical abuse towards a partner than those without a mental disorder, and the risk is higher in men than women.[11] A recent longitudinal study indicated that comorbid diagnoses of substance use and personality disorders increase the risk of intimate partner violence perpetration against women in men with mental disorders.[12] However, few studies reported on recent violence (within the past year), and there was little data on whether violence occurred during acute episodes of mental illness, or on the role of substance use as a confounder.

Evidence regarding identification of DVA perpetration in both mental health and other healthcare settings is lacking. Qualitative studies have indicated that mental health professionals have little confidence in enquiring about and responding to DVA in general, and specifically about DVA perpetration.[13 14] A medical records review of male patients in UK primary care showed that only 0.5% had DVA perpetration or victimisation documented in their medical records, despite 16% of the same patients disclosing negative behaviours in their relationship with a partner in a contemporaneous survey.[15] A systematic review has highlighted that mental health service users are in favour of routine enquiry of experiences of DVA by healthcare professionals and value when clinicians respond to disclosure by being non-discriminatory and supportive towards those who reported having experienced DVA.[6] There is a small but emerging literature exploring the experiences of DVA perpetrators through qualitative studies, but to our knowledge no review has sought to synthesise evidence on DVA perpetrators' experiences of engaging with healthcare services, or of the facilitators and barriers to disclosure of DVA perpetration to healthcare professionals.

This review aims to add to the evidence base by systematically identifying and reviewing qualitative studies that have explored the experiences of DVA perpetrators who access healthcare services, and by using meta-ethnography to synthesise and produce a new interpretation of this phenomenon. We aimed to address the following questions:

1. What are DVA perpetrators' experiences of being asked about and discussing DVA with healthcare staff?
2. What are DVA perpetrators' experiences of using healthcare services when their history of DVA perpetration is known to healthcare staff?

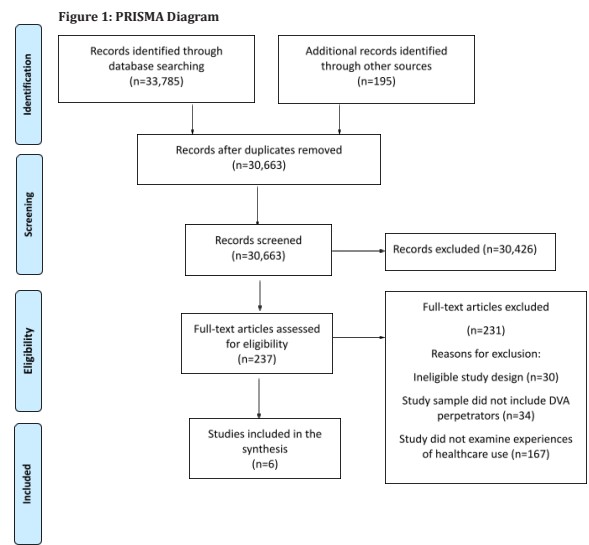

Figure 1: PRISMA Diagram

**Figure 1** Preferred Reporting Items for Systematic Reviews and Meta-Analysis diagram.

3. What are DVA perpetrators' experiences of seeking, accessing or being referred by healthcare staff for support to address their relationship difficulties and abusive behaviours?
4. What are DVA perpetrators' views on the association between their health and their relationship difficulties and abusive behaviours?

## METHODS

We used the Preferred Reporting Items for Systematic Reviews and Meta-Analysis (PRISMA)[16] and Meta-Etnography Reporting Guidelines (eMERGe) checklists[17] when writing our report. Please see figure 1 for PRISMA checklist and online supplemental appendix 1 for eMERGe checklist.

### Inclusion criteria

Studies meeting the following criteria were included:

► Population: Adults (>16 years of age) of any gender who are known to be, or identified themselves as, current or former perpetrators of DVA. DVA is defined as any incident, or pattern of incidents, of controlling, coercive or threatening behaviour, violence or abuse, between individuals 16 years or older who are, or have

been, intimate partners or family members, regardless of gender or sexuality.

► Phenomenon of interest: Experiences of using healthcare services.

► Setting: Healthcare services (primary care, emergency medicine, mental health services, secondary care medical or surgical specialties).

► Design: Any qualitative or mixed methods study design.

► Type: Studies published in peer-reviewed journals, in the grey literature (ineligible formats listed under exclusion criteria), including masters' dissertations and doctoral theses.

► Language of publication: English, Spanish, Portuguese or Italian.

► Date of publication: Published from the start date of each database until 15 March 2020.

### Exclusion criteria

► Design: Studies not using qualitative methods.

► Setting: Studies that analysed experiences of use of treatment programmes for DVA perpetrators ('perpetrator programmes'). Perpetrator programmes are interventions with specific aims that do not always involve healthcare services and have been studied separately from the wider healthcare literature.[18]

► Type: Book chapters, conference papers, editorials, letters or general comment papers.

► Language of publication: Studies published in languages other than English, Spanish, Italian or Portuguese.

► Studies in which the published data were not sufficiently detailed to allow inclusion in the metasynthesis, such as when participant accounts and/or

researchers' interpretations were not clearly stated, or when data collection and analysis methods were not reported.

No geographic restrictions were set for this review.

### Information sources

The following electronic bibliographic databases were searched: Cochrane, MEDLINE, Embase, PsycINFO, HMIC, BNID, CINAHL, ASSIA, IBSS, SSCI for peer-reviewed literature and NDLTD, OpenGrey and SCIE Online for grey literature.

Additionally, the primary reviewer conducted forwards citation tracking of included articles using Web of Science and Google Search.

### Search strategy

The search strategy used combinations of search words for DVA, IPV or FV; for qualitative research; for perpetration of violence and abuse; and for healthcare settings, to ensure that the search results matched the study questions and inclusion criteria. Please see table 1 for search terms. The databases were searched from their respective start dates to 15 March 2020.

### Patient and public involvement statement

Patients or the public were not involved in the design, or conduct, or reporting, or dissemination plans of our research.

### STUDY RECORDS
### Data management

Records were managed using EndNote software. The studies were coded using NVivo V.11 software[19] and data were extracted into MS Excel.

**Table 1** Search terms

| DVA act—OR | Qualitative research—OR | DVA perpetrator—OR |
|---|---|---|
| Intimate partner violence; Intimate partner abuse; Partner abuse OR violence<br>Spouse abuse OR violence;<br>Battered wom*;<br>Couple* abuse OR violence;<br>Domestic abuse OR violence;<br>Family abuse OR violence;<br>Coercive control;<br>Control$ behavio$r;<br>((abus$ OR batter$ OR violen$ OR beat$) adj2 (domestic OR partner$ OR family OR families OR spouse OR woman OR women OR men OR man OR female$ OR male$ OR wife OR wives OR husband$ OR boyfriend$ OR girlfriend$ OR elder$ OR brother$ OR sister$ OR father$ OR mother$ OR daughter$ OR son$)<br>(domestic adj2 homicid$).mp | Qualitative research/<br>Interview/<br>Theme$ or thematic<br>Qualitative study; Qualitative<br>Ethnograph$;<br>Narrative;<br>Account;<br>Participant observation; Grounded theory;<br>Interpretative; Phenomenological Analysis;<br>Case stud$;<br>Focus group;<br>Framework analysis; Framework approach;<br>Mixed-method$<br>Content analysis<br>Discourse analysis | Perpetrator$; Offender$; Batterer$; Abuser$<br>**Setting:**<br>Patient$ OR<br>Service user$ OR<br>Consumer of (mental)health care<br>Health$ OR<br>Healthcare<br>Service$<br>Clinic$<br>Hospital$<br>Emergenc$<br>Outpatient |

DVA, domestic violence and abuse.



## Selection process

The primary reviewer (author MAC) conducted the searches and downloaded the results of each database into a separate file on a records management software (EndNote X9). A combined file was created at the end of the search and duplicate entries were removed.

The primary reviewer screened the studies' titles and abstracts and selected those that matched the inclusion criteria, followed by the screening of full-text records. The secondary reviewer screened the titles and abstracts of a random selection of 250 abstracts and 25 full-text articles (which amounted to over 10% of full-text studies identified). Consensus regarding inclusion was reached for all records after discussion between the reviewers and the senior authors. The number of studies excluded and the reasons for exclusion of full-text studies are documented and reported in figure 1.

## Data extraction process

A data extraction form was developed by the primary reviewer (MAC) and was piloted by the two reviewers with four studies. The pilot was refined until a final data extraction form was agreed by all coauthors. The secondary reviewer performed an independent check of the data extraction in all studies included in the review.

Data were extracted on study characteristics including: sample size; participant gender; type of DVA; type of healthcare setting; and country. Using meta-ethnography methods, data were collected on first-order and second-order constructs.

## Data synthesis

Data were synthesised using meta-ethnography. This method was chosen as meta-ethnography aims to arrive at a higher order of interpretation, with the potential of going beyond a synthesis of the original studies to generate new knowledge on a topic, while maintaining the integrity of interpretations from the primary studies.

Meta-ethnography treats the views of study participants as 'first-order constructs' and the explanations of each study's authors as 'second-order constructs'. The reviewers sought to identify and synthesise the first-order and second-order constructs that were similar across studies via a process called 'reciprocal translation'.[20] The relationships that reviewers identified between findings from the different primary studies formed the 'third-order constructs'.[20] The reviewers looked for contradictory findings within and between studies (intra-study and inter-study contradictions) via the process of refutational translation, and examined whether there were plausible explanations for these apparent contradictions, such as different contextual factors between studies.[20 21]

## Meta bias(es)
### Risk of selection bias

As described in the selection process, the reviewing team included a secondary reviewer, who conducted screening of a proportion of the search results separately. This method aimed to minimise the risk of bias arising from misclassification of studies' eligibility for inclusion in the review.[22]

### Risk of reporting bias

The meta-ethnography method addresses risk of bias in the reporting of studies by considering the expressed views of participants and those of the study authors separately, as described in the Data Synthesis section.

## Confidence in cumulative evidence

We used the GRADE-CERQual (Confidence in the Evidence from Reviews of Qualitative Research) approach[23–29] to assess the confidence in the review findings.

## Risk of bias in individual studies

The studies included had their quality appraised independently by each reviewer. There was full agreement between both reviewers about the quality appraisal scores for all studies.

Quality assessment was based on Harden et al,[30] who suggest 12 criteria based on the quality of reporting, use of strategies to increase reliability and validity, and the extent to which the findings reflected participants' views and experiences. Studies were classified as high quality (those meeting 10 or more criteria out of 12), medium quality (those meeting between 7 and 9 criteria) and low quality (those meeting fewer than 7 criteria). Due to the current lack of consensus in the literature regarding quality assessment of qualitative studies,[31 32] the quality assessment was not used to exclude studies from the review. Please see online supplemental appendix 2 for the details on the quality assessment of each study.

## Reflexivity

All authors are women and based in the UK. The authors' professional backgrounds are varied and include clinical and academic psychiatry (MAC and LMH), undergraduate medical school (SB), academic medical sociology (HL) and applied health research (SO). Authors MAC and LMH also have direct experience of assessing and treating mental health service users who have been subject to, or perpetrated, DVA; MAC as a liaison psychiatrist based at a general acute hospital and LMH as a perinatal psychiatrist.

All authors believe that the evidence for the need for healthcare services to address gender-based violence is compelling, and that evidence on identifying and addressing DVA perpetration in healthcare is crucial. The lead author (MAC) explicitly considered how her professional background and previous experiences of assessing and treating DVA survivors and perpetrators may have influenced the research process. She reflected on previous clinical experiences and discussed those with her coauthors during the data collection and analysis, so as to limit the influence of professional or personal biases in the conduct of the research.

**Table 2** Study characteristics

| Study | Setting | Country | Sample size and characteristics | Community or forensic sample | Quality appraisal score (please see online supplemental appendix 2 for details) |
|---|---|---|---|---|---|
| Hester et al[37] | Voluntary perpetrator programmes | UK | 62 male IPV perpetrators | Mixed | Low (5/12) |
| van Rooij et al[33]* | Community forensic psychiatry | The Netherlands | 9 male and 1 female perpetrators of IPV and FV | Forensic | Medium (8/12) |
| Morgan et al[35]* | General practice | UK | 8 male IPV perpetrators | Community | Medium (9/12) |
| Swogger[38] | Prison | USA | 15 male IPV perpetrators | Forensic | Medium (9/12) |
| Bacchus et al[34]* | Sexual health clinic | UK | 10 male IPV perpetrators in same sex relationships† | Community | Medium (7/12) |
| Hashimoto et al[36] | Addictions services | UK | 20 male IPV perpetrators | Community | Medium (7/12) |

*These studies also included other participants who were not perpetrators of domestic violence and abuse. Data from those participants were not included in the analysis.
†These participants disclosed perpetration of at least one abusive behaviour towards a partner in a survey and/or in the qualitative interviews, but not all described themselves as 'IPV perpetrators'.
FV, family violence; IPV, intimate partner violence.

## RESULTS

A total of 237 abstracts were identified as potentially meeting the inclusion criteria during title and abstract screening. Following full-text screening (see figure 1), six studies were included in the review; four of these studies were published in peer-reviewed journals, one in a published report and one is a doctoral thesis.

Studies reported data from 125 participants who met the review criteria; all were adults and all but one of whom were males (see table 2). Other socio-demographic information, for example, ethnicity, was not included in all papers. Five of the included studies focused on IPV; only van Rooij et al[33] included perpetrators of FV as well as IPV perpetrators. One study[34] focused on men in same-sex relationships while the other five studies either focused on heterosexual relationships or did not mention the type of intimate partner relationships.

Types of healthcare setting included general practice[35]; community addictions service[36]; sexual health clinic[34] and community forensic psychiatry[33]; two studies[37 38] explored participants' experiences of healthcare without specifying the setting. Half of the included studies (n=3) recruited participants from community, non-forensic settings,[34–36] and the remaining three studies recruited perpetrators of DVA from forensic settings, including perpetrator programmes,[37] the criminal justice system[33] or prison.[38] For the latter group of studies, the constructs included in the analysis of this review related to experiences of use of healthcare services (including forensic psychiatry) but not of perpetrator programmes (see exclusion criteria). Four studies were conducted in the UK, one in the Netherlands[33] and one in North America.[38]

All studies included individual qualitative interviews as the method of data collection; some studies also collected quantitative data, which were not analysed for this review. The three studies that reported the qualitative analysis methods[33 35 38] applied grounded theory[39]; the remaining three studies did not specify the data analysis methods used. The quality of reporting was assessed as medium in five of the six studies and as low in one.

The lead author (MAC) extrated data for analysis from the full primary studies following repeated reading of each individual study; extraction was independently checked by the secondary reviewer (SB). Data were exported to NVivo for coding.[19] Coding was independently checked by coauthors SO and HL; codes and themes were discussed by the authors and refinements were made where necessary.

Data were coded according to whether they originated from direct participant quotes (first-order constructs) or from researchers' interpretations (second-order constructs); and according to the themes or subthemes they represented at the initial stage of analysis. After the initial coding stage, further rounds of coding and re-arranging of concepts and themes were conducted following discussions between the authors, until the final themes for first-order and second-order constructs were identified. Third-order constructs were identified via an iterative process of re-reading the first-order and second-order constructs and the full papers, translating the papers into each other by comparing the meaning of themes and their relationships across studies and developing an overarching model or 'line of argument synthesis' for the phenomenon studied. This was achieved by successive

**Table 3** First-order constructs

| Themes | First-order constructs |
| --- | --- |
| Emotions towards DVA perpetration | Shame, embarrassment, regret[36]<br>Feeling blamed[38] |
| Attitudes towards DVA perpetration | Normalisation, minimisation, low prioritisation of relationship problems[33 36] |
| Experiences of healthcare use | Positive experiences: good rapport with healthcare workers; feeling listened to; accessing interventions that address abusive behaviours[33 38]; practical support[33]<br>Negative experiences: feeling blamed by healthcare services[38]; couples' therapy focusing on the victim; delays and difficulties in accessing support,[36 37] believing that healthcare workers lack expertise on DVA[36] or do not know enough about service users' personal lives[34]<br>Referring to 'anger' as a way of disclosing DVA[36 37] |
| Triggers to change | Escalation of abuse; relationship breakdown[35 37]<br>Fear of loss of contact with children[37]<br>Social consequences, including being arrested or being banned from social spaces[35 37]<br>Positive engagement with healthcare services[38] |
| Attitudes towards enquiry about relationship problems | Enquiry is more acceptable at the initial assessment of an individual by a healthcare service,[34] and if it is linked with the healthcare problems for which that individual is seeking help[34 35]<br>Availability of emotional support is a necessary element to allow enquiry about relationship problems[34] |
| Concerns about confidentiality | Fear of children's social services' involvement[34 36]<br>Perception that certain services (eg, sexual health clinics) are more private than general practice[34] |

DVA, domestic violence and abuse.

discussions between the authors. The context of each study, including characteristics of population and setting, and their strengths and limitations were taken into account in the interpretations.

### First-order constructs
Analysis of the six qualitative papers identified 27 first-order constructs, which were grouped into six key themes (see table 3; participants' emotions towards DVA; attitudes towards DVA; experiences of healthcare use; triggers to wanting to instigate behavioural changes with regards to DVA perpetration; attitudes towards enquiry about DVA by healthcare services' staff; and concerns about confidentiality.

### Emotions about DVA perpetration or relationship difficulties
Some participants described experiencing shame, embarrassment and regret over their abusive behaviours towards intimate partners and reported to researchers that those emotions meant they were unwilling to admit to those behaviours to clinicians.[36] Other participants reported feeling blamed by healthcare services when disclosing DVA perpetration, particularly when they had also experienced DVA in their relationships and when the abuse led to the involvement of child protection services.[38]

### Attitudes towards DVA perpetration or relationship difficulties
Some participants minimised the importance of DVA, placing more emphasis on the positive aspects of their intimate relationship. Others normalised DVA, reporting views that abusive behaviours were an expected part of

intimate relationships, and did not require specific help or intervention,[36] indicating lack of awareness of what constitutes abuse in a relationship. Participants who reported that other issues in their lives (such as housing or financial problems) took priority over their relationship problems voiced that they were less likely to disclose relationship difficulties to healthcare professionals or to seek any type of support for those difficulties,[33 36] reflecting a low prioritisation of DVA among other stressors.

### Experiences of healthcare use
Some participants described how it took them many years to decide to seek help for DVA perpetration. Some experienced a long and convoluted route to access the support they wanted via healthcare services,[36 37] while others did not seek help from healthcare professionals as they believed healthcare staff lacked the knowledge or expertise to address the perpetration of abusive behaviours.[36] When participants decided to seek help from healthcare professionals for relationship problems, some specifically wanted a referral for an 'anger management' programme.[36 37] A range of different services were accessed, for example, addictions services, general practice and mental health services.

Experiences of psychological support were mixed, with some participants describing how individual or couples' therapy had been ineffective in reducing their use of abusive behaviours, or how they had felt blamed for their relationship problems within therapy.[38] Couples' counselling was perceived as unhelpful in changing perpetrators'

behaviour when the intervention was focused on the victim's experiences rather than on the perpetrator's experiences and behaviour.[38] Participants in the study focusing on men in same-sex relationships thought that their general practitioners did not know enough about their personal life, particularly about their sexuality, to allow disclosure of relationship problems.[34]

Participants in the studies set in prisons and in community forensic psychiatry, however, described how they were able to establish a close rapport with an individual healthcare worker who listened to them, or that they were pleased to gain access to psychotherapeutic interventions to focus on addressing their abusive behaviours.[33 38] Practical support, such as help with public transport fares to attend appointments, was appreciated. In one case, healthcare use led to access to professional reports that assisted in the participant's court case related to DVA.[33] Another attributed positive change to a combination of psychiatric treatment and a perpetrator programme undertaken while serving a prison sentence for DVA[38]; this is described in more detail in the theme below.

### Triggers that instigate behavioural changes with regards to DVA perpetration

Participants articulated how escalation of abuse, relationship breakdown, fear of losing contact with their children, being arrested or suffering other consequences of their behaviour (such as being banned from social spaces) acted as important triggers to disclosing DVA perpetration and to seeking help via healthcare services.[35 37] Some participants viewed those events as 'wake-up calls' to come to terms with their behaviour and seek help. Engagement with mental healthcare and programmes for reduction of violent behaviour offered by prison healthcare services appeared to be a catalyst for change for one participant, who reported positive changes such as increased self-awareness and feeling less overwhelmed by his emotions through treatment.[38]

### Attitudes towards enquiry about DVA by healthcare services' staff

Participants' views about the acceptability of enquiry about DVA varied. In the study set in general practice, some thought that targeting enquiry only to individuals that show signs associated with DVA (eg, physical marks or emotional distress) was more acceptable than universal enquiry in that setting[35]; nevertheless, one participant (who had disclosed perpetration of physical, psychological and verbal abuse against a partner in the research questionnaire) was completely opposed to any questions about DVA, as he perceived DVA or relationship problems as an entirely private matter.[35]

In the sexual health study, selective enquiry was also mentioned as preferable to universal enquiry by one participant; another participant was in favour of enquiry, but suggested that the initial assessment was the ideal time for it. Both participants felt that that an explanation about the link between DVA and the health problems addressed by that service would help individuals understand the rationale for, and feel more comfortable with, the question. In the same study, the provision of emotional support from a counsellor or non-medical health advisors was also mentioned as a condition for the acceptability of enquiry about DVA.[34]

### Concerns about confidentiality

Concerns about confidentiality acted as a key barrier to disclosure of relationship problems to clinicians.[34 36] Participants in the study set in an addictions clinic feared the potential consequences of disclosure, such as involvement from children's social services. One participant described that, for him, it was preferable to endure the consequences of being in a mutually abusive relationship than to seek help, due to his fear of losing the custody of his children if his relationship problems became known.[36]

Having information about DVA documented in their general practice medical records was also a concern for some; other services, such as sexual health clinics, were perceived to offer more privacy and anonymity than primary care.[34]

### Second-order constructs

Thirty second-order constructs were identified; these were grouped in nine themes. Of those, six themes were duplications of the themes that emerged from the first-order constructs (emotions towards DVA; attitudes around DVA; experiences of healthcare use; triggers to wanting to instigate behavioural changes with regards to DVA perpetration; attitudes towards enquiry about DVA by healthcare services' staff; and concerns about confidentiality); those will not be reported in this section. Three new themes emerged that had not appeared in first-order constructs: the help-seeking journey; healthcare services' ability to address DVA; and under-reporting of relationship difficulties (see table 4).

### Help-seeking journey

Researchers in the study set in perpetrator programmes described how participants accessed healthcare services through different routes: a reactive route, following contact with other services (often the police or social services) due to their abusive behaviour, or a proactive route, when individuals sought help directly through healthcare for their relationship problems or a health problem related to their abusive behaviours. Researchers interpreted that individuals who sought help proactively and were explicit about their use of abusive behaviours found the help they desired more easily. Usually this was in the form of gaining access to a treatment programme for DVA perpetrators or to another intervention directly addressing abusive behaviour.[37]

Some individuals sought help for their relationship problems in indirect ways, by reporting anger[36 37] or common mental health problems such as depressive symptoms.[37] These participants reported finding it unhelpful when doctors responded to this type of help-seeking by prescribing antidepressant medication or



| Table 4 | Second-order constructs |
|---|---|
| **Themes** | **Second-order constructs** |
| Help-seeking journey | Reactive vs proactive help-seeking[37]<br>Implicit vs explicit help-seeking[36 37] |
| Healthcare services' ability to address DVA | Healthcare services' time constraints and competing priorities[34 36]<br>Intimate partner violence 'not a legitimate issue' to discuss with healthcare staff[36]<br>Lack of relevant expertise by staff[34 36] |
| Under-reporting of relationship difficulties | Minimisation of DVA[36 38]<br>Fear of consequences[34 36]<br>Shame regarding DVA[36]<br>Lack of recognition of what constitutes DVA[34 36] |

DVA, domestic violence and abuse.

referring service users to general counselling services without exploring the reasons behind their symptoms.

Some participants hoped to access specific support to reduce abusive behaviours, often framed as 'anger management services', and were frustrated when those services proved difficult to find or to access. Researchers proposed that the ability to access services promptly was crucial, as individuals were often ambivalent towards behavioural change and might have changed their minds about help-seeking if there was a long wait to access a service.[37]

### Healthcare services' ability to address DVA

Researchers suggested that participants' willingness to discuss relationship difficulties with staff in the addictions and sexual health settings were affected by what they perceived to be limitations of healthcare services.[34 36]

In the addictions setting, researchers identified that participants were aware of time constraints within services and decided to use their consultations to discuss problems other than their relationship difficulties. Professionals' lack of expertise to address relationship problems was also mentioned by addiction service users as a barrier to disclosure, though it was not clear to the researchers how that perception had developed, and whether it had originated in participants' beliefs about the appropriateness of discussing relationship problems in addictions services. It was noted by the researchers that many participants reported that they had never been asked about their relationships by clinicians in that setting.[36]

Concerns about clinicians' lack of time or skills to address relationship difficulties were also identified by researchers in the study located in a sexual health clinic: participants suggested that the sexual health clinic was a more appropriate venue for inquiry and disclosure of relationship problems than primary care, which was perceived as a setting where staff did not have the skill or time to explore issues about domestic abuse. Sexual health clinics were also seen by participants in that study as a more 'private' setting where sensitive and intimate issues could be discussed. Researchers recommended training for staff in healthcare services to increase awareness of

DVA and improve their communication skills to help build trusting relationships with service users.[34]

### Under-reporting of relationship difficulties

Some participants under-reported DVA perpetration; researchers' interpretations of that included lack of recognition of what constitutes DVA,[34] minimisation,[36 38] shame and fear of the consequences of disclosure,[34] including loss of contact with one's children through social services intervention.[36] In two studies, participants denied having ever perpetrated abuse, despite having been convicted of DVA-related offences or having answered affirmatively to questions about DVA perpetration in the quantitative survey in the same study.[36 38] Researchers concluded that the denial or lack of awareness indicated a mismatch between actions (even those that participants had admitted to) and perception of what constitutes DVA. This may have contributed to the tendency towards under-reporting of DVA perpetration to healthcare professionals[34 36] in addition to factors such as shame and fear of the adverse consequences of disclosure.

### Third-order constructs

The constructs identified in the 6 studies were suited for the creation of a *line or argument synthesis*, in which studies identify different aspects of a phenomenon and can be integrated to provide a new interpretation. Other possible approaches under the meta-ethnography method are reciprocal synthesis or refutational synthesis. Although certain themes were present in a number of studies, during analysis it became evident that the studies approached different aspects of DVA perpetrators' healthcare experiences which complemented one another, making a line of argument synthesis more appropriate than reciprocal synthesis. There were no major or unexplained contradictions within studies or between studies, making the data not suitable for a refutational synthesis.

The third-order constructs originated from this meta-ethnography can be divided in factors that facilitate DVA perpetrators' disclosure of relationship difficulties to healthcare staff and their engagement with healthcare

| Table 5 | Third-order constructs |
|---|---|
| **Third-order construct** | **Description** |
| Facilitators of disclosure of DVA to healthcare staff and engagement with healthcare | Reaching a crisis point or experiencing negative social consequences following abusive behaviour<br>Active listening by healthcare professionals<br>Availability of emotional and practical support (ideally on-site) |
| Barriers to disclosure of DVA to healthcare staff and engagement with healthcare | Negative emotions and attitudes towards DVA by perpetrators<br>Lack of recognition of what constitutes DVA<br>Fear of consequences of disclosure<br>Lack of trust in healthcare services' knowledge or expertise in addressing DVA |

DVA, domestic violence and abuse.

services; and factors that act as barriers to disclosure or DVA or healthcare service engagement (see table 5).

### Factors that facilitate DVA perpetrators' disclosure of DVA to healthcare staff and help-seeking

Some individuals reported having explicitly requested help to reduce abusive behaviours, in some cases triggered by a crisis (such as separation from a partner, escalation of abuse or arrest), and sought or expected a specific intervention to address their problems. Others hinted about abuse when speaking to clinicians, referring to 'relationship problems', 'anger' or reporting low mood. Active listening by professionals, further exploration of relationship history, skills in enquiring about relationship difficulties sensitively and linking these to service users' health concerns tended to facilitate engagement of DVA perpetrators with healthcare services. The availability of on-site support for relationship problems may also act as a facilitator and may be a necessary condition for some to disclose their relationship problems. It was unclear what form this immediate support needs to take beyond non-judgmental listening by professionals or practical support.

### Factors that act as barriers to disclosure of DVA and engagement with healthcare services

The studies demonstrated that willingness to disclose relationship difficulties to healthcare staff is negatively affected by perpetrators' perceptions that healthcare professionals lack expertise to address DVA, and by their own low prioritisation of relationship problems.

Some individuals feel shame, guilt or regret when discussing abusive behaviours with healthcare staff, whereas others see abusive behaviours as a normal part of an intimate relationship, or do not recognise their own behaviours as abusive. These factors affect whether (and how) individuals seek help. Concerns about being blamed for relationship problems and about confidentiality, particularly due to fears of involvement from other services (eg, children's social services), also act as important barriers to disclosure of DVA to healthcare professionals. These concerns, combined with negative emotions and attitudes towards DVA, are some of the factors underlying the under-reporting of DVA by perpetrators.

### Line of argument synthesis

The third-order constructs identified in this review have been integrated into a proposed model for how perpetrators of DVA experience and engage with healthcare services. DVA perpetration is a complex phenomenon that can be experienced differently by individuals, and many perpetrators do not identify as such. An intra-study contradiction was identified involving the constructs of attitudes and emotions related to DVA. Some participants expressed shame and regret over their behaviours, while others minimised or normalised DVA.[36] The minimisation and normalisation are consistent with the findings of a recent qualitative study of male IPV perpetrators in treatment for addiction and their female partners, which examined the perspective of perpetrators and their partners on the role of substance use in IPV.[40] In that study, perpetrators tended to attribute IPV to isolated events triggered by specific disputes, while their partners described a pattern characterised by enduring abuse, including severe violence and coercive control in some cases. The intra-study contradiction identified in our review highlights how individual emotional reactions to DVA by perpetrators can vary, with emotions ranging from denial and normalisation to self-reflection, shame and regret over their actions.

The journey to disclosure of perpetration of abusive behaviours to healthcare services is heterogeneous and can start with an individual moving from a position of not identifying themselves as a perpetrator of DVA, to accepting that label due to reaching a point of crisis with wider social consequences, such as being banned from certain spaces or being in contact with the criminal justice system. This, however, is not the case for all individuals who perpetrate DVA; some will remain unwilling to disclose abusive behaviours to healthcare staff due to strong negative emotions and attitudes towards DVA perpetration, including denial, minimisation and possibly a lack of understanding of what constitutes abuse.

When individuals had already experienced the adverse consequences of perpetrating abusive behaviours, such as relationship separation, escalation of abuse or exclusion from social spaces, those events acted as triggers to disclosure and help-seeking. In these situations, perpetrators had decided to disclose their difficulties to clinicians



to seek help and avoid further negative consequences. Those who were able to be more explicit about their problems to healthcare staff had a more positive experience of the support received, indicating that openness about the use of abusive behaviours and ability to discuss them with a healthcare professional in an environment perceived as non-judgmental can be the start of a positive engagement with healthcare.

From the perspective of healthcare services, there are challenges in engaging with DVA perpetrators due to the barriers outlined above, as well as the perception by some individuals that healthcare services are not the right place to discuss DVA, or lack the expertise to understand the problem and provide the necessary support. Healthcare services have to tread a fine line between engaging individuals using a non-judgmental approach while also maintaining the safety of others, including children, which will often require breaking patient confidentiality. Encouragingly, some DVA perpetrators think that interventions by healthcare staff, such as providing active listening and emotional or practical support, can help, though it was not clear what specific support was desired beyond referrals to interventions such as anger management programmes.

### Confidence in review findings: GRADE-CERQual assessment

The GRADE-CERQual assessment was applied to all included studies, as shown in table 6. The assessment identified that all studies were relevant to the review questions and populations. Four out of six studies included only male perpetrators of IPV in heterosexual (or not otherwise specified) relationships, while one study included perpetrators of IPV and FV (including one woman) and another study included only male perpetrators of IPV in same-sex relationships. All of these are relevant to the review questions, though the implications of their specific findings will be discussed later in this paper. Minor-to-moderate concerns with regards to methodological limitations were identified, due to studies not reporting the methods used to establish the validity of data collection tools or data analysis methods, and/or issues with quantity or richness of data in study findings; none of the findings presented a high level of concern. Due to the small number of studies and participants, we were unable to assign a high level of confidence to any of the findings, although there was a good level of coherence between study findings (considering the small study numbers) with each third-order construct being anchored by at least one-third of the included studies.

### DISCUSSION
### Main findings

Our findings indicate that the experiences of healthcare use by DVA perpetrators are complex, and may depend on whether individuals self-identify as perpetrators of abusive behaviours and on what stage they are in their help-seeking journey. When healthcare services provide a listening space and offer emotional and practical support to those who are ready to indicate (even if tentatively) that there are problems in their relationships, individuals feel more inclined to disclose abusive behaviours to clinicians.

The barriers to the disclosure of abusive behaviours by perpetrators to healthcare workers include feelings of shame, beliefs that abuse is a normal part of intimate relationships (or a lack of recognition of what constitutes abuse), concerns about confidentiality and lack of trust in healthcare professionals' ability or readiness to address DVA. The latter finding, in particular, is consistent with the experiences of healthcare professionals regarding their readiness to address DVA in their practice, as identified by a recent systematic review.[41]

Overall, it is important to recognise that healthcare services need to navigate the experiences and preferences of DVA perpetrators while maintaining safe professional practice and safeguarding survivors of abuse. One key barrier to disclosure of perpetration of abusive behaviours was the concern about confidentiality and the desire to avoid negative social consequences, such as the perceived threat of losing contact with one's children via involvement of children's social services.[36] Given the risk posed by DVA to children[8–10] and criticism made of services regarding poor inter-agency working and information-sharing in cases of domestic homicides,[42] it would be inappropriate and unsafe for healthcare staff not to share information pertaining to the physical or emotional safety of children with other agencies such as social services or (when appropriate) police. It is, therefore, a challenge for healthcare services to establish ways in which to provide individuals with a listening space while being clear about the limits of confidentiality.

The study also found that some negative consequences of DVA perpetration, such as involvement of children's social services, being banned from social spaces, or separation may act as barriers or facilitators to disclosure to healthcare services. This apparent contradiction can be explained by the context of the perpetrators' experiences and the timing of help-seeking, as outlined in the previous section. Importantly, some of these factors, such as separation, are known to be risk factors for domestic homicides,[42 43] and are crucial for the assessment of risk by healthcare professionals.

In common with our findings, a recent systematic review on the healthcare experiences of DVA survivors[44] found that a positive relationship with a clinician and a safe and confidential environment acted as facilitators to disclosure of DVA, while fear of the consequences of disclosure and concerns that healthcare services may not be able to help acted as barriers to disclosure. Lack of recognition what constitutes abuse, or the normalisation of abuse have also been reported by studies conducted with survivors. However, a host of other themes identified in the survivors' studies, including concerns about personal safety, feelings of powerlessness, financial dependence on the perpetrator, lack of social support or avoidance of re-living trauma, were absent from studies involving

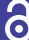

**Table 6** GRADE-CERQual assessment

| Review findings | Studies contributing | Methodological limitations | Relevance | Coherence | Adequacy of data | CERQual assessment |
|---|---|---|---|---|---|---|
| Reaching a crisis point or experiencing negative social consequences following abusive behaviour | Morgan et al[35]; Hester et al[37] | Minor concerns regarding data collection and analysis (Morgan et al[35]). Moderate concerns regarding reporting of recruitment/sampling, data collection and analysis (Hester et al[37]) | Relevant (both studies) | Two out of six studies. There was one inter-study contradiction explained by the context of the perpetrators' experiences and the timing of help-seeking. | Minor concerns over quantity (Morgan et al[35]) and richness of data (Morgan et al[35], Hester et al[37]) | Moderate confidence (minor concerns over adequacy of data and methodological limitations) |
| Active listening by healthcare professionals | van Rooij et al[33]; Swogger[38] | Minor concerns regarding data collection and analysis (both studies) | Relevant (both studies) | Two out of six studies | Moderate concerns over richness (van Rooij et al[33]) and quantity of data (Swogger et al[38], van Rooij et al[33]) | Moderate confidence (moderate concerns over adequacy of data and minor concerns over methodological limitations) |
| Availability of emotional and practical support (ideally on-site) | van Rooij et al[33]; Swogger[38], Bacchus et al[34] | Minor concerns regarding data collection and analysis (all three studies) | Relevant (all three studies) | Three out of six studies | Moderate concerns regarding quantity of data (Swogger et al[38], van Rooij et al[33]). Minor concerns over richness of data (Bacchus et al[34]) | Moderate confidence (moderate concerns over adequacy of data and minor concerns over methodological limitations) |
| Negative emotions and attitudes towards DVA by perpetrators | van Rooij et al[33]; Hashimoto et al[36]; Swogger[38] | Minor concerns regarding data collection and analysis (all three studies) | Relevant (all three studies) | Three out of six studies. There was one intra-study contradiction explained by variation in individual emotional reactions to DVA by perpetrators | Moderate concerns regarding quantity (Swogger[38], van Rooij et al[33]) and richness of data (van Rooij et al[33]). No concerns (Hashimoto et al[36]) | Moderate confidence (moderate concerns over adequacy of data in two out of three studies, and minor concerns over methodological limitations in all three studies) |
| Fear of consequences of disclosure | Hashimoto et al[36]; Bacchus et al[34] | Minor concerns regarding data collection and analysis (both studies) | Relevant (both studies) | Two out of six studies | No concerns (Hashimoto et al[36]). Moderate concerns regarding quantity and richness of data (Bacchus et al[34]) | Moderate confidence (moderate concerns over adequacy of data in one study, and minor concerns over methodological limitations in both studies) |
| Lack of trust in healthcare services' knowledge or expertise in addressing DVA | Hashimoto et al[36]; Bacchus et al[34], Swogger[38] | Minor concerns regarding data collection and analysis (all three studies) | Relevant (all three studies) | Three out of six studies | Minor concerns over quantity of data (all three studies) | Moderate confidence (minor concerns over adequacy of data and methodological limitations) |

CERQual, Confidence in the Evidence from Reviews of Qualitative Research; DVA, domestic violence and abuse; GRADE, Grading of Recommendations Assessment, Development and Evaluation.



perpetrators, demonstrating the vulnerability of survivors and the power imbalances characteristic of DVA.[45]

All but one of the participants in the studies included were men, therefore some issues identified may be linked to gendered patterns of help-seeking. It has been reported that men have lower rates of help-seeking for health problems,[46] and a systematic review has described a number of barriers to help-seeking for health problems in men;[47] some of these barriers were also present in our findings, such as embarrassment, fear and difficulty communicating with healthcare professionals. Traditional masculine ideologies may also be a factor in barriers to perpetrators of DVA, or more specifically IPV, engaging with healthcare; a longitudinal study of male IPV perpetrators who used substance use services found that participants who espoused traditional masculine ideologies were less likely to engage with supporting services such as healthcare and employment or vocational support.[48]

### Strengths and limitations of this review
Our review provides the first systematic review and synthesis of qualitative research exploring the healthcare experiences of DVA perpetrators, addressing a significant gap in the current literature on DVA in healthcare services. We used a rigorous systematic review methodology, including a comprehensive systematic search and the GRADE-CERQual approach to assess the confidence in each review finding.[49]

The a priori review questions were mostly answered: DVA perpetrators' experiences of discussing DVA with healthcare staff; their experiences of using healthcare services when their history of DVA perpetration is known; and their experiences of seeking support to address relationship difficulties were explored in the original studies and in the interpretive findings of this review. However, the review question regarding DVA perpetrators' views on the association between their health and relationship difficulties or abusive behaviours could not be addressed, as it was not explored in any of the contributing studies.

One of the limitations of the review is the small number of studies found; twice as many studies were included in a meta-synthesis of the experiences of DVA survivors,[50] indicating that DVA perpetration continues to be an under-researched topic relative to experiences of DVA. Most studies, with the exception of Hester *et al*[37] had a relatively small number of participants. It is important to recognise that, as well as being under-researched, the DVA perpetrator population may also be hard to reach, due to its low identification in healthcare services; additionally, there are challenges, including ethical ones, in recruiting participants from this group. The healthcare experiences of DVA perpetrators were not the primary focus of most included studies and made up only a small proportion of the data presented. There was sufficient data to identify common themes between the studies and develop a 'line of argument' synthesis, but the availability of data limited the authors' ability to develop a new theory regarding the topic.

Studies included in the review were conducted in a variety of healthcare settings; this is a strength, in that the representation of different healthcare settings and clinical populations has helped address a variety of aspects of participants' healthcare experiences. However, this affects the generalisability of certain findings. One study, for example, included only individuals in same-sex relationships,[34] whereas the other five studies either did not specify the type of relationship or included only individuals in heterosexual relationships[35–38] (or a mixture of intimate partner relationships and adult family relationships[33]). Certain findings, such as the importance of privacy in a healthcare encounter, were identified only in the study conducted with individuals in same-sex relationships, suggesting that stigma leading to identity concealment and other types of 'minority stress' associated with being in a same-sex relationship[51 52] may add an additional barrier to disclosure of DVA perpetration.

Another aspect of this review that is both a strength and a limitation is the fact that the majority of study participants were male perpetrators of IPV. This limits how the review findings can be generalised to the (significantly under-researched) population of perpetrators of FV, but it is a strength in that we can have higher confidence in the findings in relation to male IPV perpetrators.

This review did not identify any studies from non-Western, non-English speaking or low-income and middle-income countries that met inclusion criteria, nor studies focusing specifically on perpetrators of non-partner adult FV or female perpetrators of DVA. This raises the question of whether the findings are generalisable to those populations.

### Implications for clinical practice and policy
Our findings demonstrate some of the complexities associated with addressing DVA perpetration in healthcare services. Relationship difficulties and DVA can elicit a variety of reactions in perpetrators, some of which may be conflicting, and which affect the likelihood of disclosure of abusive behaviours to healthcare professionals. Additionally, individuals may present to healthcare services with the aim to seek help for the consequences of relationship difficulties (eg, low mood) without disclosing DVA perpetration directly. This highlights the importance of a trusting rapport between healthcare professionals and service users, as well as willingness on the part of clinicians to explore the intimate and family relationships of their service users in more detail.

Some of the participants of the studies included in this review articulated that DVA was not a legitimate issue to bring to consultations with healthcare professionals,[36] while others stated that it seemed appropriate to discuss DVA with clinicians in services that were perceived as more 'private', such as sexual health services.[34] An assurance of privacy may be particularly important for individuals in same-sex relationships, as previously discussed. Moreover, some participants seemed unaware of what behaviours constitute abuse.[34 36] This suggests that, beyond raising

awareness of DVA perpetration among healthcare professionals, it may be necessary to raise public awareness of what behaviours constitute perpetration of abuse. Public awareness campaigns on DVA, including the campaign by the UK Home Office during the COVID-19 pandemic,[53] tend to address survivors or those at risk of experiencing DVA; campaigns directed at perpetrators will likely need a different language and will need to be circulated through different channels to address the populations at risk of perpetrating DVA. For those campaigns to be effective, the training of health professionals (and others who are likely to encounter DVA, such as the police and social care) on how to safely enquire about DVA perpetration and how to respond to disclosures will be essential.

A number of participants reported that they expected to receive support from healthcare services to access interventions to reduce their anger or violence ('anger management'), or a treatment programme for DVA perpetrators. Although a systematic review of qualitative studies has demonstrated that users of IPV perpetrator programmes find value in those interventions,[54] the evidence for effectiveness of DVA perpetrator programmes in reducing risk to survivors is limited[55–57]; most intervention studies are restricted to specific populations or settings, making them less generalisable, and many lack a control group. However, psychological therapies such as behavioural couples' therapy or cognitive behavioural therapy combined with treatment for alcohol misuse can be effective in reducing abusive behaviours.[57–60] This suggests that the existing role of healthcare services in identifying and treating substance use may provide a route into addressing DVA perpetration when those issues coexist.

## CONCLUSION

Our results provide the first synthesis of evidence on the healthcare experiences of DVA perpetrators. This meta-synthesis indicates that healthcare professionals need to be open to asking questions about service users' families and intimate relationships, and be attentive to requests for help that may not be explicit. These findings can inform the development of training to improve the identification of DVA perpetration in healthcare settings. Our findings also suggest a possible role for public health campaigns directed at groups at risk of DVA perpetration to improve awareness of what constitutes abuse and encourage individuals to disclose those behaviours to healthcare professionals. DVA perpetration is a complex and multifactorial issue that requires the involvement of a number of sectors, and further research is needed in how best to integrate the responses of healthcare with those of other agencies, such as social care, police and criminal justice system.

**Contributors** MAC: conceptualising the study; drafting and finalising review protocol; systematic literature search; primary reviewer for study selection and appraisal; meta-synthesis; wrote first draft of study for publication. SB: secondary

reviewer for study selection and appraisal. LMH, HL, SO: conceptualising the study; advising on methodology; supervision and editing of drafts.

**Funding** MAC is a self-funded MD (Res) student and received salary support through a 3-month full-time secondment to the Section of Women's Mental Health, Health Service and Population Research Department, IoPPN, KCL, from July–September 2018, from an NIHR Research Professorship to LH (NIHR-RP-R3-12-011). LH is an NIHR Senior Investigator. The funding organisations had no involvement in the study design; in the collection, analysis and interpretation of the data; in the writing of the report and in the decision to submit the study for publication.

**Competing interests** All authors have completed the ICMJE uniform disclosure form at www.icmje.org/coi_disclosure.pdf and declare: LH and SO receive salary support from the UKRI Cross-disciplinary Network on Violence, Abuse and Mental Health (www.vamhn.co.uk). LH also receives salary support from the NIHR SLAM/KCL Biomedical Research Centre and the NIHR South London Applied Research Collaboration. MAC received salary support through a 3-month full-time secondment to the Section of Women's Mental Health, Health Service and Population Research Department, IoPPN, KCL, from July–September 2018, from an NIHR Research Professorship to LH (NIHR-RP-R3-12-011). HL and SB declare no financial relationships with any organisations that might have an interest in the submitted work in the previous three years. The authors declare no other relationships or activities that could appear to have influenced the submitted work.

**Patient consent for publication** Not required.

**Ethics approval** No ethical approval was sought as this study is a systematic review that analysed anonymised data from published studies, which have obtained informed consent/ethical approval.

**Provenance and peer review** Not commissioned; externally peer reviewed.

**Data availability statement** Data are available in a public, open access repository. The study involved analysis of anonymised participant data available in published research studies, a published report and an unpublished doctoral thesis. All studies included are referenced in the paper and are available online through research databases. DOIs or weblinks for the included studies are provided below. van Rooij FB, Ten Haaf J, Verhoeff AP. Temporary restraining orders in the Netherlands: a qualitative examination of perpetrator and victim views. Journal of Family Violence. 2013 Jul 1;28(5):503-14. https://doi.org/10.1007/s10896-013-9520-2 https://doi.org/10.1007/s10896-013-9520-2. Morgan K, Williamson E, Hester M, et al. Asking men about domestic violence and abuse in a family medicine context: Help seeking and views on the general practitioner role. Aggression and violent behavior. 2014 Nov 1;19(6):637-42. https://doi.org/10.1016/j.avb.2014.09.008. Hashimoto N, Radcliffe P, Gilchrist G. Help-seeking behaviours for intimate partner violence perpetration by men receiving substance use treatment: a mixed-methods secondary analysis. Journal of interpersonal violence. 2018 May 1:0886260518770645. https://doi.org/10.1177/0886260518770645. Bacchus LJ, Buller AM, Ferrari G, et al. 'It's Always Good to Ask': a mixed methods study on the perceived role of sexual health practitioners asking gay and bisexual men about experiences of domestic violence and abuse. Journal of Mixed Methods Research. 2018 Apr;12(2):221-43. https://doi.org/10.1177/1558689816651808. Hester M, Westmarland N, Gangoli G, et al. Domestic violence perpetrators: identifying needs to inform early intervention. Newcastle: Northern Rock Foundation. 2006. https://aura.antioch.edu/etds/303/. Swogger R. Incarcerated Men and the Etiology of Intimate Partner Violence (PhD thesis). Antioch University. 2016. http://aura.antioch.edu/cgi/viewcontent.cgi?article=1309&context=etds.



**ORCID iD**
Marilia A Calcia http://orcid.org/0000-0001-8488-9342

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
