## [Reviewer comments · BMJ Open]

ARTICLE DETAILS

TITLE (PROVISIONAL)	The healthcare experiences of perpetrators of domestic violence and abuse: a systematic review and meta-synthesis
AUTHORS	Calcia, Marilia; Bedi, Simran; Howard, Louise; Lempp, Heidi; Oram, Sian

VERSION 1 – REVIEW

REVIEWER	Hegarty, Kelsey University of Melbourne, Dept of General Practice
REVIEW RETURNED	25-Aug-2020

GENERAL COMMENTS	This is a great synthesis of an under researched area in the field of domestic violence. Early engagement through the health system of perpetrators is essential in the secondary prevention of domestic violence. It is not a surprise so few studies were found but this paper does outline the lack of investigation into the views of help-seeking of perpetrators of DV. The abstract is clear. The introduction is UK focussed which might need to be expanded for an international journal such as BMJ Open e.g. use WHO definition and add global prevalence. References need to be updated e.g. 3 is extremely old and should use systematic reviews e.g. WHO. The methods are clear but did the authors consider using CERQUAL to assess quality of the studies (see https://www.cerqual.org/). Perhaps there is a need to make a statement why not? The results are interesting but it became clear that some of the men were discussing heterosexual relationships and others same-sex relationships. This point was not really raised to any great extent but it has influenced the findings. I think one of the studies was from a sexual health clinic and this heterogeneity of settings may limit the findings. This needs to be a limitation but also referenced in the themes found. The discussion is not referenced to any great extent particularly the conclusion. What struck me is how the findings are similar to victims' facilitators and barriers e.g. women don't recognise abuse as abuse and would benefit from community campaigns. Highlighting what is particular about men would be good in the discussion. Limitations need to say more how this form of analyses may not be good for such a small number of studies. Minor points include: reword p21 line 6 trusting rapport sentence check references as some look to be the wrong formatting.
--

REVIEWER	Childress, Saltanat The University of Texas at Arlington
-----------------	---

REVIEW RETURNED	26-Oct-2020
-------------

GENERAL COMMENTS	This paper focuses on an under researched population, perpetrators of domestic violence and abuse, and specifically on what is known from qualitative studies of perpetrators' disclosure within healthcare settings. The paper addresses an important area of research about understanding perpetrators' disclosure and how they may become connected to interventions. The conceptual constructs that are brought out about the experiences of disclosure (i.e., barriers and facilitators to disclosure) are interesting and useful. However, there are several concerns about the narrowness of the conceptualization of the problem and the narrowness of the conceptualization of the meta-review and the search. The first concern is that this review is only based on six articles. The reader begins to wonder what is the value-added to the field of a review when the literature is this limited and this thin. The kind of context from the broader review of the universe of studies which might strengthen the rationale for the narrow focus is not provided. The second concern is the scope itself is too narrow by restricting the inclusion criteria only on healthcare settings. One can see the potential logic of aiming this review at healthcare professionals, but because of the narrowness, the analysis presents a very small aperture through which to consider the meaningfulness of the results. Why not include other type of settings, particularly, law enforcement or other social services, or even couple-oriented literature? A review with this large of an initial universe of studies could shed light on the quantity of literature on perpetrators' disclosure in general to contextualize the subset of articles on healthcare settings. With such a narrow and under-researched area as male perpetrators' disclosure, narrowing to only healthcare settings without justifying the selection based on a wider contextualization limits the value of the review. The third concern is the review only includes qualitative articles. The question is if this is such a small field of study to begin with, why to focus only on qualitative articles? If there are also quantitative, why not also include quantitative studies in a systematic review of this type? The review does not explain if there are any, but that would be interesting to know. It would be interesting to know what a quantitative study of this nature might find, if in fact any exist. In terms of the specific (1st, 2nd, and 3d order) constructs, they seem to be well identified to talk about selection of themes, and there is some good material there about what qualitatively happens in disclosure. All of these are interesting main findings, although their robustness is an admitted limitation. This article would benefit from clarifying and motivating more strongly the rationale for the inclusion criteria and providing a stronger contextualization in the literature for the selected focus. Providing some context about the literature on perpetrator disclosure in general, and more sense of the scope of the literature on perpetrator disclosure in, for example, law enforcement, and social services, the justification for the narrow focus on qualitative perpetrator disclosure in healthcare settings might become stronger and more useful to the field. The tightly zoomed in view on this thin slice of the literature without more context about the status of the literature on male perpetrator disclosure limits the usefulness of the study as a guide for research and intervention.
---

REVIEWER	Tarzia, Laura University of Melbourne, Department of General Practice
-----------------	--

GENERAL COMMENTS

Thanks for the opportunity to review this manuscript, which makes a critically important contribution to the field. To my knowledge, no previous studies have synthesised the qualitative literature on perpetrators' healthcare experiences. Yet, understanding the experiences and expectations of this population is vital to developing appropriate responses and avenues for early engagement.

I enjoyed reading the manuscript and thought it was extremely well-written. My main comments are based around the methodology and framing of the findings:

1. Overall, the methodology section lacks detail. In particular there was little information provided about the analysis and the use of meta-ethnography. I think it is important to justify why meta-ethnography was chosen as opposed to other methods. It also would have been good to use one of the newer reporting guidelines for meta-ethnographies (see France et al. 2019) to strengthen this section of the paper.
2. Reflexivity was not really addressed anywhere that I could see - please add some brief detail on how the authors' perspectives may have influenced the analysis.
3. I was a little confused about the research questions outlined on page 4-5. These did not seem to be fully addressed by the findings. I understand that there may not have been enough data in the included studies to answer all the questions you set out to address in the beginning, however, it would be good to acknowledge this in the discussion section perhaps.
4. It was a bit disappointing that only random selections of studies were reviewed by a second reviewer during the screening process. In particular I wondered why, when there were only 6 included full-text studies, the second reviewer looked at only two of them?!
5. Despite undertaking a quality appraisal of the studies, there appeared to be no further mention of the generally low level of quality of the evidence. With no studies ranked higher than "Medium" it seems to me that the findings can only be used with caution (in terms of making any recommendations for practice). Obviously your findings also demonstrate that there is a real lack of methodologically sound research in this area!
6. Although I realise that this review is the first of its kind and we know little about this topic, I would have liked to see third-order constructs beyond "barriers and facilitators". There was a real opportunity here to think creatively about how perpetrators engage with healthcare systems and to generate some new theory that might offer insights into improvements in practice. It seems to me that meta-ethnography requires a bit more interpretation than what has been demonstrated here. There is nothing wrong with synthesising barriers and facilitators, but the review was set up as something broader and more complex than that (health care experiences).

In summary, I think this review would be greatly improved with some tightening up between the aims/RQs and the findings, more

	detail around the meta-ethnographic methods and perhaps some reframing of the third-order constructs. Otherwise, I congratulate the authors on an important contribution. Minor points:  - The second dot point under Exclusion Criteria says "Phenomenon of interest" but I don't think this is what it is addressing?? - An additional limitation is the small number of participants in many of the included studies, which may impact on the strength of the findings.
--	--

VERSION 1 – AUTHOR RESPONSE

Reviewer: 1

3. The abstract is clear. The introduction is UK focussed which might need to be expanded for an international journal such as BMJ Open e.g. use WHO definition and add global prevalence. References need to be updated e.g., 3 is extremely old and should use systematic reviews e.g., WHO.

We have updated our introduction (page 3) to include global WHO statistics, as recommended by our reviewer. We have retained the UK definition of domestic violence in this paragraph, however, as this was the definition used by the review. The relevant text now reads as follows:

“Domestic violence and abuse (DVA) are highly prevalent worldwide. According to the World Health Organisation (WHO), 27% of women aged 15-49 years who have been in an intimate relationship have experienced physical or sexual intimate partner violence (IPV), and 38% of female victims of homicide have been killed by an intimate partner. In the UK, the definition of DVA includes family violence as well as IPV. DVA is defined in the UK Domestic Abuse Bill 2020 as abusive behaviour (which may be physically, sexually, emotionally, psychologically or economically abusive or violent, threatening, controlling or coercive) by one person to another, when both are aged 16 or over and are “personally connected”.”

4. The methods are clear but did the authors consider using CERQUAL to assess quality of the studies. Perhaps there is a need to make a statement why not?

Following our reviewer’s recommendation, we have conducted a CERQUAL assessment and report on confidence in the review findings. We report on page 9 that: “We have used the GRADE-CERQual (Confidence in the Evidence from Reviews of Qualitative Research) approach to assess the confidence in the review findings for third-order constructs.”

On page 22, we report that: “The GRADE-CERQual assessment was applied to included studies, as shown in Table 6. The assessment identified that all studies included were relevant to the review questions and populations. Four out of 6 studies included only male perpetrators of IPV in heterosexual relationships, while one study included perpetrators of IPV and FV (including one female) and another study included only male perpetrators of IPV in same-sex relationships. All of these are relevant to the review questions, though the implications of their specific findings will be discussed later in this paper. Minor to moderate concerns with regards to methodological limitations were identified, due to studies not reporting the methods used to establish the validity of data collection tools, data analysis methods, and/or quantity or richness of data in most study findings; no findings presented a high level of concern. Due to the small number of studies and participants, we

were unable to assign a high level of confidence to any of the findings, although there was a good level of coherence between study findings (considering the small study numbers) with each third-order construct being anchored by at least one third of the included studies.”

Study level assessments are presented in Table 6, on pages 22-25.

We also now report on page 27 that “We used a rigorous systematic review methodology, including comprehensive systematic search and used the GRADE-CERQual to assess the confidence in each review finding.”

5. The results are interesting but it became clear that some of the men were discussing heterosexual relationships and others same-sex relationships. This point was not really raised to any great extent but it has influenced the findings. I think one of the studies was from a sexual health clinic and this heterogeneity of settings may limit the findings. This needs to be a limitation but also referenced in the themes found.

We agree that it is important to discuss this point, and now do so within our strengths and limitations section on pages 27-28:

“Studies included in the review were conducted in a variety of healthcare settings; this is a strength, in that the representation of different healthcare settings and clinical populations has helped address a variety of aspects of participants’ healthcare experiences. However, this affects the generalizability of certain findings. One study, for example, included only individuals in same-sex relationships, whereas the other five studies either did not specify the type of relationship or included only individuals in heterosexual relationships (or a mixture of intimate partner relationships and adult family relationships). Certain findings, such as the importance of privacy in a healthcare encounter, were identified only in the study conducted with individuals in same-sex relationships, suggesting that stigma leading to identity concealment and other types of ‘minority stress’ associated with being in a same-sex relationship may add an additional barrier to disclosure of DVA perpetration.”

6. The discussion is not referenced to any great extent particularly the conclusion. What struck me is how the findings are similar to victims’ facilitators and barriers e.g. women don’t recognise abuse as abuse and would benefit from community campaigns. Highlighting what is particular about men would be good in the discussion.

We thank our reviewer for this suggestion, and address this point on pages 26-27, as follows:

"In common with our findings, a recent systematic review on the healthcare experiences of DVA survivors found that a positive relationship with a clinician and a safe and confidential environment acted as facilitators to disclosure of DVA, while fear of the consequences of disclosure and concerns that healthcare services may not be able to help acted as barriers to disclosure. Lack of recognition what constitutes abuse, or the normalisation of abuse have also been reported by studies conducted with survivors. However, a host of other themes identified in the survivors’ studies, including concerns about personal safety, feelings of powerlessness, financial dependence on the perpetrator, lack of social support, or avoidance of re-living trauma, were absent from studies involving perpetrators, demonstrating the vulnerability of survivors and the power imbalances characteristic of DVA.

All but one of the participants in the studies included were male, therefore some issues identified may be linked to patterns of help-seeking among men. It has been reported that men have lower rates of help-seeking for health problems, and a systematic review has described a number of barriers to help-seeking for health problems in men, some of these barriers were also present in our findings, such as embarrassment, fear and difficulty communicating with healthcare professionals. Traditional

masculine ideologies may also be a factor in barriers to perpetrators of DVA, or more specifically IPV, to engaging with healthcare; a longitudinal study of male IPV perpetrators who used substance use services found that participants who espoused traditional masculine ideologies were less likely to engage with supporting services such as healthcare and employment or vocational support."

7. Limitations need to say more how this form of analyses may not be good for such a small number of studies.

Due to the small number of included studies, we were not able to generate theory from the review findings; we were however able to identify common themes. We now state this clearly within our strengths and limitations section (page 27), as follows:

"There was sufficient data to identify common themes between the studies and develop a 'line of argument' synthesis, but the availability of data limited the authors' ability to develop a new theory regarding the topic."

8. Reword p21 line 6 trusting rapport sentence

Thank you; we have corrected this.

9. Check references as some look to be the wrong formatting.

Thank you; we have updated this.

Reviewer: 2

Dr. Saltanat Childress, The University of Texas at Arlington

10. The first concern is that this review is only based on six articles. The reader begins to wonder what is the value-added to the field of a review when the literature is this limited and this thin. The kind of context from the broader review of the universe of studies which might strengthen the rationale for the narrow focus is not provided.

We thank reviewer 2 for their comments. As outlined in our Introduction, there is a small but emerging body of research on the topic of DVA perpetrators' engagement with healthcare services. As per the comments by Reviewers 1 and 3, evidence on how to promote early engagement of perpetrators through healthcare services is crucial in addressing DVA, and a synthesis of these studies is needed to address the current concerns about the lack of effective identification of this population. Despite the small number of studies (which was discussed in the limitations, see response to point 7, above), we were able to identify common themes and emerging recommendations and highlights gaps in the literature to guide future research.

11. The second concern is the scope itself is too narrow by restricting the inclusion criteria only on healthcare settings. One can see the potential logic of aiming this review at healthcare professionals, but because of the narrowness, the analysis presents a very small aperture through which to consider the meaningfulness of the results. Why not include other type of settings, particularly, law enforcement or other social services, or even couple-oriented literature? A review with this large of an initial universe of studies could shed light on the quantity of literature on perpetrators' disclosure in general to contextualize the subset of articles on healthcare settings. With such a narrow and under-researched area as male perpetrators' disclosure, narrowing to only healthcare settings without justifying the selection based on a wider contextualization limits the value of the review.

We appreciate Reviewer 2's concerns about the narrow focus of the review. However, and as

highlighted by the World Health Organisation (WHO 2013; 2021) and National Institute of Healthcare Excellence (NICE, 2014), healthcare providers are in a unique position to identify and respond to perpetrators of DVA. Healthcare settings are fundamentally different from social care, police, or criminal justice system settings in how they operate and how they can identify and address DVA. Our focus on healthcare settings was, therefore, intentional and designed to make specific findings and recommendations that can be useful for healthcare services.

12. The third concern is the review only includes qualitative articles. The question is if this is such a small field of study to begin with, why to focus only on qualitative articles? If there are also quantitative, why not also include quantitative studies in a systematic review of this type? The review does not explain if there are any, but that would be interesting to know. It would be interesting to know what a quantitative study of this nature might find, if in fact any exist.

We are glad to provide clarification on this point. Previous research identified that health professionals have low confidence in addressing DVA (Oram et al; 2016; Rose et al 2011), and Domestic Homicide Reviews have identified a problem of perpetrators' poor engagement of healthcare services (UK Home Office, 2013). Given this context, we specifically sought to explore the experiences of DVA perpetrators when accessing healthcare services, particularly around enquiry, disclosure, and accessing support. Those questions could not have been adequately answered by studies using only quantitative methods and these studies were therefore excluded from the review.

13. In terms of the specific (1st, 2nd, and 3d order) constructs, they seem to be well identified to talk about selection of themes, and there is some good material there about what qualitatively happens in disclosure. All of these are interesting main findings, although their robustness is an admitted limitation.

We thank the reviewer for highlighting that the themes are well identified. As detailed under point 4, above, we have now conducted additional appraisal of review findings using the GRADE-CERQual tool and comment specifically on our confidence in each review finding. As now reported on page 22, "no findings presented a high level of concern" and "there was a good level of coherence between study findings (considering the small study numbers) with each third-order construct being anchored by at least one third of the included studies." As we also acknowledge on page 22, however, "due to the small number of studies and participants, we were unable to assign a high level of confidence to any of the findings"

14. This article would benefit from clarifying and motivating more strongly the rationale for the inclusion criteria and providing a stronger contextualization in the literature for the selected focus. Providing some context about the literature on perpetrator disclosure in general, and more sense of the scope of the literature on perpetrator disclosure in, for example, law enforcement, and social services, the justification for the narrow focus on qualitative perpetrator disclosure in healthcare settings might become stronger and more useful to the field. The tightly zoomed in view on this thin slice of the literature without more context about the status of the literature on male perpetrator disclosure limits the usefulness of the study as a guide for research and intervention.

We thank the reviewer for this summary of their concerns. We hope we have adequately responded to their point regarding the review scope under point 11. We have revised the manuscript to add more information on the rationale for the review focus. For example, on page 4-5, we now report that: "Evidence regarding identification of DVA perpetration in both mental health and other healthcare settings is lacking. Qualitative studies have indicated that mental health professionals have little confidence in enquiring about and responding to DVA in general, and specifically about DVA perpetration" and that "There is a small but emerging literature exploring the experiences of DVA perpetrators through qualitative studies, but to our knowledge no review has sought to synthesise

evidence on DVA perpetrators' experiences of engaging with healthcare services or of the facilitators and barriers to disclosure of DVA perpetration to healthcare professionals.”

Reviewer: 3

Dr. Laura Tarzia, University of Melbourne

Comments to the Author:

15. Overall, the methodology section lacks detail. In particular there was little information provided about the analysis and the use of meta-ethnography. I think it is important to justify why meta-ethnography was chosen as opposed to other methods. It also would have been good to use one of the newer reporting guidelines for meta-ethnographies (see France et al. 2019) to strengthen this section of the paper.

We are glad to provide further justification for our analytical approach, as suggested by our reviewer, and to use the eMERGE checklist to strengthen our reporting of our methods. We now report on page 8 of the manuscript that:

“Data were synthesised using meta-ethnography. This method was chosen as meta-ethnography aims to arrive at a higher order of interpretation, with the potential of going beyond a synthesis of the original studies to generate new knowledge on a topic, while maintaining the integrity of interpretations from the primary studies.”

We report on page 5 of the manuscript that:

“We used the PRISMA reporting checklist when writing our report and the eMERGE checklist for reporting of meta-ethnographies (Please see Figure 1 for PRISMA checklist and Appendix 1 for eMERGE checklist).”

We added additional detail throughout the Methods section in line with the requirements of the eMERGE checklist and provide the completed checklist in an appendix.

16. Reflexivity was not really addressed anywhere that I could see - please add some brief detail on how the authors' perspectives may have influenced the analysis.

We are glad to include this additional detail, and have added the following paragraph to our manuscript (pages 9-10):

“All authors are female and based in the UK. The authors' professional backgrounds are varied and include clinical and academic psychiatry (MAC and LMH), undergraduate medical school (SB), academic medical sociology (HL) and applied health research (SO). Authors MAC and LMH also have direct experience of assessing and treating mental health service users who have been subject to, or perpetrated, DVA; MAC as a liaison psychiatrist based at a general acute hospital and LMH as a perinatal psychiatrist.

All authors believe that the evidence for the need for healthcare services to address gender-based violence is compelling, and that evidence on identifying and addressing DVA perpetration in healthcare is crucial. The lead author (MAC) explicitly considered how her professional background and previous experiences of assessing and treating DVA perpetrators may have influenced the research process. She reflected on previous clinical experiences and discussed those with her co-authors during the data collection and analysis, so as to limit the influence of professional or personal biases in the conduct of the research. ”

17. I was a little confused about the research questions outlined on page 4-5. These did not seem to

be fully addressed by the findings. I understand that there may not have been enough data in the included studies to answer all the questions you set out to address in the beginning, however, it would be good to acknowledge this in the discussion section perhaps.

We agree this important to highlight, and now acknowledge this within our strengths and limitations section, on page 27:

"The a priori review questions were mostly answered: findings were able to address DVA perpetrators' experiences of discussing DVA with healthcare staff; their experiences of using healthcare services when their history of DVA perpetration is known; and their experiences of seeking support to address relationship difficulties were explored in the original studies and in the interpretive findings of this review. However, the review question regarding DVA perpetrators' views on the association between their health and relationship difficulties or abusive behaviours could not be addressed as it was not explored in any of the contributing studies.

14. It was a bit disappointing that only random selections of studies were reviewed by a second reviewer during the screening process. In particular I wondered why, when there were only 6 included full-text studies, the second reviewer looked at only two of them?!

We are glad to provide further clarification on this point. Partial double screening was conducted, as is common practice in systematic reviews. At the title and abstract stage, a second reviewer independently screened 250 records. Due to the high level of agreement between reviewers, no further independent screening was undertaken at this stage. At the full-text stage, the second reviewer screened 25 full text records; this was equivalent to over 10% of full text studies identified. Again a high level of agreement was reached, and no further independent screening was undertaken. We now report on page 8 that:

"The primary reviewer screened the studies' titles and abstracts and selected the studies that matched the inclusion criteria, followed by the screening of full-text records. The second reviewer screened the titles and abstracts of a random selection of 250 records and 25 full text records (which amounted to over 10% of full text studies identified). Consensus regarding inclusion was reached for all records after discussion between the reviewers and the senior authors. The number of studies excluded and the reasons for exclusion of full text studies is documented and reported in Figure 1."

Six studies were included in the review; independent checks of data extraction were undertaken for two of the six papers. At the recommendation of the reviewer, we have undertaken further checks: data extraction has now been checked for all included papers.

15. Despite undertaking a quality appraisal of the studies, there appeared to be no further mention of the generally low level of quality of the evidence. With no studies ranked higher than "Medium" it seems to me that the findings can only be used with caution (in terms of making any recommendations for practice). Obviously, your findings also demonstrate that there is a real lack of methodologically sound research in this area!

Thank you. As also discussed in our response to point 4, above, we have augmented our quality assessment through the use of the GRADE-CERQual tool. We report both the methodological quality of the contributing studies and our confidence in study findings. For example, on page 21, we report that:

"Minor to moderate concerns with regards to methodological limitations were identified, due to studies not reporting the methods used to establish the validity of data collection tools, data analysis methods, and/or quantity or richness of data in most study findings; no findings presented a high level

of concern. Due to the small number of studies and participants, we were unable to assign a high level of confidence to any of the findings, although there was a good level of coherence between study findings (considering the small study numbers) with each third-order construct being anchored by at least one third of the included studies.”

16. Although I realise that this review is the first of its kind and we know little about this topic, I would have liked to see third-order constructs beyond "barriers and facilitators". There was a real opportunity here to think creatively about how perpetrators engage with healthcare systems and to generate some new theory that might offer insights into improvements in practice. It seems to me that meta-ethnography requires a bit more interpretation than what has been demonstrated here. There is nothing wrong with synthesising barriers and facilitators, but the review was set up as something broader and more complex than that (health care experiences).

While our synthesis makes an important contribution to the field of research on DVA perpetration, due to the small number of studies identified, we were cautious not to over-interpret the available data. We fully agree there is a need to develop theory in this area, and we hope the findings of this review will help direct future research accordingly. We have provided further detail to the line of argument synthesis, which we hope addresses the reviewer 3's concerns. The text on pages 21-22 now reads as follows:

“The third-order constructs identified in this review have been integrated to propose a model for how perpetrators of DVA experience and engage with healthcare services. DVA perpetration is a complex phenomenon that can be experienced differently by individuals, and many perpetrators do not identify as such. An intra-study contradiction was identified involving the constructs of attitudes and emotions related to DVA. Some participants expressed shame and regret over their behaviours, while others minimised or normalised DVA. The minimization and normalisation are consistent with the findings of a recent qualitative study of male IPV perpetrators in treatment for addiction and their female partners, which examined the perspective of perpetrators and their partners on the role of substance use in IPV. In that study, perpetrators tended to attribute IPV to isolated events triggered by specific disputes, while their partners described a pattern characterized by enduring abuse, including severe violence and coercive control in some cases. The intra-study contradiction identified in our review highlights how individual emotional reactions to DVA by perpetrators can vary, with emotions ranging from denial and normalisation to self-reflection, shame and regret over their actions.

The journey to disclosure of perpetration of abusive behaviours to healthcare services is heterogenous and can start with an individual moving from a position of not identifying themselves as a perpetrator of DVA, to accepting that label due to reaching a point of crisis with wider social consequences such as being banned from certain spaces or being sentenced by the criminal justice system. This, however, is not the case for all individuals who perpetrate DVA; some will remain unwilling to disclose abusive behaviours to healthcare staff due to strong negative emotions and attitudes towards DVA perpetration, including denial, minimization and possibly a lack of understanding of what constitutes abuse.

When individuals had already experienced the adverse consequences of perpetrating abusive behaviours, such as relationship separation, escalation of abuse, or exclusion from social spaces, those events acted as triggers to disclosure and help-seeking. In these situations, perpetrators had decided to disclose their difficulties to clinicians to seek help and avoid further negative consequences. Those who were able to be more explicit about their problems to healthcare staff had a more positive experience of the support received, indicating that openness about the use of abusive behaviours and ability to discuss them with a healthcare professional in an environment perceived as non-judgmental can be the start of a positive engagement with healthcare services.

From the perspective of healthcare services, there are challenges in engaging with DVA perpetrators due to the barriers outlined above, and due to the perception by some individuals that healthcare services are not the right place to discuss DVA, or lack the expertise to understand the problem and provide the necessary support. Healthcare services have to tread a fine line between engaging individuals using a non-judgmental approach while also maintaining the safety of others, including children, which will often require breaking patient confidentiality. Encouragingly, some DVA perpetrators think that interventions by healthcare staff, such as providing active listening and emotional or practical support, can help, though it was not clear what specific support was desired beyond referrals to interventions such as anger management programmes.”

17. The second dot point under Exclusion Criteria says "Phenomenon of interest" but I don't think this is what it is addressing??

Thank you; we have now corrected this.

18. An additional limitation is the small number of participants in many of the included studies, which may impact on the strength of the findings.

We are glad to acknowledge this and now report on page 27 under strengths and limitations that:

“Most studies, with the exception of Hester et al., 2006 had a relatively small number of participants. It is important to recognize that, as well as being under-researched, the DVA perpetrator population may also be hard to reach due to its under-identification in healthcare services, and there are challenges, including ethical ones, in recruiting such populations. The healthcare experiences of DVA perpetrators were not the primary focus of most included studies and made up only a small proportion of the data presented. There was sufficient data to identify common themes between the studies and develop a ‘line of argument’ synthesis, but the availability of data limited the authors’ ability to develop a new theory regarding the topic.”

In addition to the points made by our reviewers, we corrected an error in the previous manuscript regarding the total number of participants; this was corrected from n=122 to n=125.